# COVID-19 Pandemic and the Crisis of Health Systems: The Experience of the Apulia Cancer Network and of the Comprehensive Cancer Center Istituto Tumori “Giovanni Paolo II” of Bari

**DOI:** 10.3390/ijerph17082763

**Published:** 2020-04-16

**Authors:** Nicola Silvestris, Antonio Moschetta, Angelo Paradiso, Antonio Delvino

**Affiliations:** 1IRCCS Istituto Tumori “Giovanni Paolo II” of Bari, 70124 Bari, Italy; 2Department of Biomedical Sciences and Human Oncology, University of Bari “Aldo Moro”, 70124 Bari, Italy; 3Inter-Institutional Department of Oncology, Apulian Cancer Network, 70124 Bari, Italy; antonio.moschetta@uniba.it; 4Department of Interdisciplinary Medicine, University of Bari “Aldo Moro”, 70124 Bari, Italy; 5Scientific Direction-IRCCS Istituto Tumori “Giovanni Paolo II” of Bari, 70124 Bari, Italy; a.paradiso@oncologico.bari.it; 6General Direction-IRCCS Istituto Tumori “Giovanni Paolo II” of Bari, 70124 Bari, Italy; a.delvino@libero.it

**Keywords:** COVID-19, pandemic, health system, cancer patient

## Abstract

On 11 March 2020, the World Health Organization declared a new disease caused by a novel virus characterized by rapid human-to-human transmission and named severe acute respiratory syndrome coronoavirus-2 (SARS-CoV-2) a pandemic. In terms of this ongoing international scenario, we report the situation in Apulia, a region of southern Italy that, as of April 2, has not yet been overwhelmed by this health emergency. In particular, we consider the care models that have been adopted, especially those that manage the requests of cancer patients.

Nothing will be the same again—or, at least, nothing should be the same again. The storm triggered by the COVID-19 pandemic has forced each of us into an unfamiliar, life-threatening situation, with health systems, even in the most advanced countries, in crisis [1]. In this report, we would like to focus on the avoidable deaths—in other words, the price that we pay when health systems are unprepared to manage this type of emergency. The goal is not to highlight individual responsibilities, which will have to be assessed in any case, at least from an ethical (and, consequently, political) point of view when this emergency is over. The idea is to balance the importance of various evident shortcomings (i.e., the absence of plans for pandemic management, the serious lack of personal protective equipment, and the absence of differentiated routes for non-infected patients who must have access to non-extendable diagnostic and therapeutic areas). Overall, we have verified, with dramatic evidence, that money is not everything (there have been times when the price of a mask was next to invaluable!). Europe has lacked a homogeneous economic and financial strategy at a time when it is necessary to ensure maximum osmosis between various countries and a coherent flow of information that is capable of promptly and effectively inducing radical changes in social behavior, a strategy which was apparently able to contain the contagion in China [2]. Furthermore, one could speculate that the actual societal model, based on the absolute power of the economy and on the induction of ephemeral needs, does not favor the prompt adaptation of habits that are common to the majority of citizens in order to combat this pandemic emergency.

Within this international scenario, what is the situation in Apulia, a region in southern Italy that, as of April 2, has not yet been overwhelmed by this health emergency? What care models have been adopted, especially in order to manage the requests of cancer patients? In Apulia, a regional cancer network has been set up, which coordinates the regional assistance and research activities in oncology, and which identifies the Comprehensive Cancer Center Istituto Tumori “Giovanni Paolo II” of Bari as the coordinating center. The aim of our regional cancer network is to defend the capacity of the system in continuing the treatment of cancer patients, who constitute the most relevant example of vulnerability during this COVID-19 emergency. 

The following clinical situations have been identified:

(a) Diagnostic services that can be performed on an outpatient basis: the oncologist must evaluate and assign the patient an emergency code; this code, assigned by the oncologist at his discretion, concerns the non-referability of the diagnostic/therapeutic procedure and defines a time within which this procedure must be performed; the patient, equipped with a surgical mask, must undergo epidemiological triage and the measurement of body temperature and oxygen saturation index; health workers must be equipped with appropriate personal protective equipment;

(b) Chemotherapy and radiotherapy services: the same measures as above must be taken; in the event that a center is unable to cope because of the positive COVID-19 status of healthcare workers, patients must be referred to the nearest center; the decision is referred to the Director of the Oncology Department, who is responsible for the specific geographic area. Due to the extent of the emergency, the use of oral therapies and subcutaneous administration must be favored (therapies are to be carried out at home after evaluating the clinical conditions and by using tools);

(c) Hospitalization in the medical field: this must be limited to specific cases, with daytime hospital visits being preferred whenever possible; in this phase of the pandemic, hospital admission for patients who are candidates for supportive therapy alone is excluded and patients are directed to hospices. For other cases (patients who are candidates for invasive diagnostic procedures, complex chemotherapy treatments, patients with comorbidities, candidates for antiblastic therapies, and those with iatrogenic toxicities are admitted as they would be in an ordinary regime); the preliminary clinical evaluation remains an essential element; in suspect cases, a naso-pharyngeal swab should be carried out; if the decision to hospitalize results in an admission to the emergency room, the patient must be assisted using all the precautions indicated for positive COVID-19 cases, until clinical evaluations and/or swabs exclude positivity;

(d) Hospitalization in the surgical field: COVID-free hospitals must be identified for each of the macro-areas into which the regional network is divided; the regional hospital network is organized by the regional health authorities, identifying the hospitals designated to treat positive COVID-19 patients; all other hospitals will have to try to protect their patients from contagion and diagnostics targeting recent/ongoing infections (RNA in nasal swab) will be performed before surgery. Hospital stays are shortened, with early patient discharge and surgical recovery being partially carried out at home.

Under the supervision of the directors of the Oncology Departments and with the cooperation of the different coordinators, the surgical teams are integrated with members from the different hospitals, according to a criterion of competence, and possibly resorting to rotation. Indeed, one of the lessons learned during this emergency is that treatment and therapy should be patient-centered, with surgeons moving to the hospital setting depending on patient characteristics and not vice versa. All those present in the operating room must wear, at the very least, a FFP2 mask, a superimposed surgical mask with a valve and pneumatic goggles.

With regards to the specific measures adopted in the Cancer Institute of Bari, the following measures were taken:The timely closure of all common areas (bars, canteens, chapels) and the suspension of scientific meetings or seminars;The adoption of a filter system at the entrance, for all patients, consisting of a pre-triage telephone system (accessed via the patient’s private mobile phone) that collects information on the medical service required, on the risk of contagion and on the presence of any COVID-19 symptoms; for this purpose, a tent provided by the Civil Protection Authority was set up outside the Institute to enable health workers to carry out a short interview, measure body temperature and the percentage of oxygen saturation; access to accompanying persons and visitors is prohibited (an exception is made for non-autonomous patients);The adoption of teleconferencing systems for multidisciplinary team meetings and scientific and assistance exchanges;The adoption, and timely updating, of the risk assessment documents (documents which include risks and prevention measures for health and safety in the workplace; they are mandatory for all workplaces, including hospitals (the latest version of the document approved by the IRCCS Cancer Institute of Bari for the management of the COVID-19 pandemic is shown in the Appendix A));The establishment of a commission overseen by the directors of the clinical departments, under the supervision of an authoritative professor of the University of Bari, as an external member; this committee, meeting twice a week via a teleconference, assesses the proposals for hospitalization on a case by case basis, authorizing access for patients for whom a diagnostic or therapeutic delay can worsen the prognosis (to date, Apulia has identified 11 hospitals with departments dedicated to the care of positive COVID-19 patients; two centers are active in the province of Bari (Policlinic, Faculty of Medicine University of Bari and Miulli Hospital in Acquaviva delle Fonti).

There is one final, albeit equally important, factor to be taken into consideration in this complex scenario. Never before has it been necessary for scientific authorities to be solely responsible for appraising innovative proposals and guaranteeing the objective assessment of ideas. At the moment, the common desire is clearly to find a miracle drug or a vaccine that is able, in a short time, to cure and, above all, prevent this disease, allowing us to resume our normal lives. There are parallels between this scenario and the old dream of identifying the “magic bullet” that is able to hit and kill cancer cells [3]. Recent progress made in the knowledge and treatment of malignancies has made it clear to each clinician and researcher that there are no drugs or miraculous solutions. We have learnt that small and constant steps forward are possible and were accomplished thanks to the ability to build networks, produce scientific research and share knowledge [4]. COVID-19 is an unknown virus that we are chasing in its race for globalization. Political and religious institutions continue to remind us that we are all "on the same boat" and that only common economic and health policies can help us overcome this outbreak. Researchers are also called upon to face this global challenge. How can they? By not losing awareness of the health and economic emergencies, we must remember that only preclinical and clinical research can end this typhoon, as it did in the treatment of neoplasms. We must collect not only clinical data, but also biological material, blood and tissue samples whenever possible. International health institutions are calling for the centralization and coordination of global biological and data banks. If it is true that no one can save themselves on their own, it is equally true that no one will identify the solution to this crisis without a critical multidisciplinary approach. 

In conclusion, the Apulian Cancer Network and the Cancer Institute of Bari are trying to mount a strong response to the needs of cancer patients during the COVID-19 pandemic by making available the skills of clinicians and researchers in the region, with an awareness that we are all learning. However, future generations will never forgive us if, during this shocking experience, we did not draw up immediate and comprehensive operational plans to fill the gaps and avoid preventable deaths.

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
