# Peer review of "COVID-19 Pandemic and the Crisis of Health Systems: The Experience of the Apulia Cancer Network and of the Comprehensive Cancer Center Istituto Tumori “Giovanni Paolo II” of Bari"

_ijerph, 2020, doi:10.3390/ijerph17082763_

Round 1

Reviewer 1 Report

AUTHORS

Manuscript title: COVID-19 pandemia and the crisis of health systems: the experience of the Apulia Cancer Network and of the Comprehensive Cancer Center Istituto Tumori “Giovanni Paolo II” of Bari

A short but meaningful manuscript, well written and straightforward. I advise publication after addressing minor concerns.

Would suggest changing the title from pandemia to pandemic, as pandemia is the Greek expression, rarely in use in the English language, hence having a broader acceptance in an international audience

Line 31: “apparently able to contain the contagion in China. “ can be referenced.

Line 45: what is the basis for the emergency code? Which are the categories and ow are they evaluated? This would feed valuable data to the reader

Line 59: How is a COVID-free hospital identified? What are the prerrogatives?

Are diagnostics implemented targeting recent/ongoing infection (IgM in serum or RNA in nasal swab?)? 

Are diagnostics done before surgery to reduce risk?

Are hospital stays shortened? Surgical recoveries can partially be made from home, was this foreseen?

Author Response

Thank you for the encouraging words.

We sincerely thank the reviewer and we are sure that her/his suggestions have helped us improve our manuscript.

Would suggest changing the title from pandemia to pandemic, as pandemia is the Greek expression, rarely in use in the English language, hence having a broader acceptance in an international audience

Line 31: “apparently able to contain the contagion in China. “ can be referenced.

We add the reference

Line 45: what is the basis for the emergency code? Which are the categories and how are they evaluated? This would feed valuable data to the reader

This code, attributed by the oncologist at his discretion, concerns the non-referability of the diagnostic / therapeutic procedure and defines a time within which this procedure must be performed

Line 59: How is a COVID-free hospital identified? What are the prerogatives?

Are diagnostics implemented targeting recent/ongoing infection (IgM in serum or RNA in nasal swab?)?

Are diagnostics done before surgery to reduce risk?

Are hospital stays shortened? Surgical recoveries can partially be made from home, was this foreseen?

The regional hospital network is organized by the regional health authorities, identifying the hospitals designated to treat positive COVID-19 patients; all other hospitals will have to try to protect their patients from contagion Diagnostic targeting recent/ongoing infection (RNA in nasal swab) will be performed before surgery. Hospital stays are shortened with early discharges of the patient and surgical recovery partially carried out at home.

Reviewer 2 Report

Dear authors,

Thanks for sharing your ideas. The ideas are good in theory but do you have any experiences from real-life situations regarding the bullet points a-d? In orders to faciliate for others to learn from your ideas it might be usuful with very detailed examples on how you work precitally with the bullet Points. Bullet a- the oncologist must assign an emergency code - what kind of code is that? how does it differ from other situations?  Bullet c)- how does it differ from usual care and why? the most dangerous thing with the covid19 is missed regular diseases. How do you manage such situation?   Do you have any data on any bullet Point? or at least identified problems with the bullet Points?

Author Response

Thank you for the encouraging words.

We sincerely thank the reviewer and we are sure that her/his suggestions have helped us improve our manuscript

Thanks for sharing your ideas. The ideas are good in theory but do you have any experiences from real-life situations regarding the bullet points a-d? In order to facilitate for others to learn from your ideas it might be useful with very detailed examples on how you work precitally with the bullet Points.

Bullet a- the oncologist must assign an emergency code - what kind of code is that? how does it differ from other situations? 

This code, attributed by the oncologist at its discretion, concerns the non-referability of the diagnostic / therapeutic procedure and defines a time within which this procedure must be performed

Bullet c)- how does it differ from usual care and why? the most dangerous thing with the covid19 is missed regular diseases. How do you manage such situation?  

In this phase of the pandemic, hospital admission for patients who are candidates for supportive therapy alone are excluded and are directly addressed to hospices. For the other cases (patients candidate for invasive diagnostic procedures, complex chemotherapy treatments, patients with comorbidities,  candidates for antiblastic therapies, and with iatrogenic toxicities are admitted on the basis of an ordinary regime) Do you have any data on any bullet Point? or at least identified problems with the bullet Points?

We are collecting all the data and the critical issues that are manifesting themselves in the realization phase even if it is too early to draw conclusions